# Neuronal let-7b-5p acts through the Hippo-YAP pathway in neonatal encephalopathy

Vennila Ponnusamy [1,2], Richard T. H. Ip[3], Moumin A. E. K. Mohamed[3], Paul Clarke [4,5], Eva Wozniak[6], Charles Mein[6], Leslie Schwendimann[7], Akif Barlas[8], Philippa Chisholm[9], Ela Chakkarapani[10], Adina T. Michael-Titus[3], Pierre Gressens [7,11], Ping K. Yip [3,12✉] & Divyen K. Shah [3,8,12]

Despite increasing knowledge on microRNAs, their role in the pathogenesis of neonatal encephalopathy remains to be elucidated. Herein, we identify let-7b-5p as a significant microRNA in neonates with moderate to severe encephalopathy from dried blood spots using next generation sequencing. Validation studies using Reverse Transcription and quantitative Polymerase Chain Reaction on 45 neonates showed that let-7b-5p expression was increased on day 1 in neonates with moderate to severe encephalopathy with unfavourable outcome when compared to those with mild encephalopathy. Mechanistic studies performed on glucose deprived cell cultures and the cerebral cortex of two animal models of perinatal brain injury, namely hypoxic-ischaemic and intrauterine inflammation models confirm that let-7b-5p is associated with the apoptotic Hippo pathway. Significant reduction in neuronal let-7b-5p expression corresponded with activated Hippo pathway, with increased neuronal/nuclear ratio of Yes Associated Protein (YAP) and increased neuronal cleaved caspase-3 expression in both animal models. Similar results were noted for let-7b-5p and YAP expression in glucose-deprived cell cultures. Reduced nuclear YAP with decreased intracellular let-7b-5p correlated with neuronal apoptosis in conditions of metabolic stress. This finding of the Hippo-YAP association with let-7b needs validation in larger cohorts to further our knowledge on let-7b-5p as a biomarker for neonatal encephalopathy.

[1] Ashford and St. Peter's Hospitals NHS Foundation Trust, Chertsey, UK. [2] Centre for Genomics and Child Health, Blizard Institute, Barts and The London School of Medicine and Dentistry, Queen Mary University of London, London, UK. [3] Centre for Neuroscience, Surgery and Trauma, Blizard Institute, Barts and The London School of Medicine and Dentistry, Queen Mary University of London, London, UK. [4] Norfolk and Norwich University Hospitals NHS Foundation Trust, Norwich, UK. [5] Norwich Medical School, University of East Anglia, Norwich, UK. [6] Genome Centre, Barts and the London School of Medicine and Dentistry, London, UK. [7] Université de Paris, NeuroDiderot, Inserm, 75019 Paris, France. [8] The Royal London Hospital, Barts Health NHS Trust, London, UK. [9] Homerton University Hospital NHS Foundation Trust, London, UK. [10] Translational Health Sciences, Bristol Medical School, University of Bristol, Bristol, UK. [11] Centre for the Developing Brain, Kings College London, London, UK. [12] These authors jointly supervised this work: Ping K Yip, Divyen K Shah. ✉email: p.yip@qmul.ac.uk

Neonatal encephalopathy (NE) such as that associated with perinatal hypoxia-ischaemia remains an important cause of morbidity and mortality in term-born neonates globally[1]. Mild therapeutic hypothermia (TH) has been shown to reduce death or severe neurodisability with a number needed to treat of seven to nine[2,3]. Improved understanding of the pathophysiology of NE is important in identifying reliable biomarkers and more effective treatments. Potentially microRNAs may have an important role in the pathophysiology of NE[4].

MicroRNAs (miRNAs), a group of small noncoding RNAs made up of 17–22 nucleotides, are important regulators of a number of processes in health and disease, in conditions including cancer[5], cardiovascular disease[6], inflammatory disorders[7] and immature brain injury[4]. There are relatively few studies on the role of miRNAs in NE, using neonatal blood samples[8,9] and umbilical cord samples[10,11]. Hence in the present study, we set out to identify candidate miRNAs from blood samples of newborns with NE and to investigate the corresponding cellular and molecular mechanisms linked to these miRNAs through the use of in vitro and in vivo models of NE.

In our analysis of miRNA next-generation sequencing (NGS) of neonates with NE, the apoptotic Hippo signalling pathway was identified to be associated with a number of miRNAs that were differentially expressed between the favourable and unfavourable outcome groups, in particular the miRNA let-7b-5p. The Hippo signalling pathway is an important, well-conserved pathway known to function across a number of mammalian species[12,13] with a role in regulating apoptosis and cell proliferation. Within this pathway, there are kinase complexes that negatively regulate the activities of the mammalian Yorkie (YKi) homologue named Yes Associated Protein (YAP) and transcriptional coactivator with PDZ-binding motif (TAZ).

The function of the Hippo pathway is dependent on the translocation of YAP/TAZ from the cytoplasm into the nucleus. In physiological conditions, the Hippo pathway is not activated so the YAP/TAZ complex remains unphosphorylated and is translocated into the nucleus. In the nucleus, YAP/TAZ bind to Transcription enhancer factor-1 (TEA/TEF) domain transcription factors 1–4 (TEAD 1–4), which are key DNA-binding platforms for YAP/TAZ to regulate cell growth in humans[14]. The nuclear YAP/TAZ/TEAD protein complex transcribes anti-apoptotic genes and maintains cell survival[15]. However, in pathological conditions such as stress, activation of the Hippo pathway leads to activation of YAP through phosphorylation. The phosphorylated YAP/TAZ remains in the cytoplasm and undergoes proteolytic degradation. In our study, we show using both in vitro and in vivo models of NE that neuronal apoptosis is associated with a decrease in let-7b-5p and a reduced nuclear location of YAP. This indicates a potential role of the Hippo pathway in the cell death associated with NE.

## Results

### Identification of candidate and endogenous control miRNAs in moderate to severe neonatal encephalopathy using miRNA next-generation sequencing.

From the four cohorts of neonates (1A: moderate to severe NE with TH and favourable outcome, 1B: moderate to severe NE with TH and unfavourable outcome, 2: mild NE without TH, 3: healthy cord blood, $n = 4$ per group) a total of 2795 miRNAs were identified in the miRNA NGS analysis performed using dried blood spots (DBS). Unsupervised data analysis using principal component analysis (PCA) did not reveal any unique clustering based on any of the simple parameters (i.e. time, injury or outcome) (Fig. 1a). A number of miRNAs were differentially expressed between the groups with unadjusted $p$ value < 0.05 (Fig. 1b) and the heatmap generated after correcting

for false discovery rate (FDR) < 0.05 revealed 17 significant miRNAs (Fig. 1c) which were differentially expressed in healthy cord blood compared to that of neonatal blood samples. On removing umbilical cord blood samples from the analysis, and correcting for FDR, there were no miRNAs that had a significant differential expression between the three remaining groups (Fig. 1d) as demonstrated in the heatmap (Fig. 1e).

Among the neonates with NE, there was a pool of 134 significant miRNAs differentially expressed between the mild NE and moderate to severe NE with favourable and unfavourable outcome groups. Of these, (top 10 in descending order of significance), miR-29b-3p, miR-3200-3p, let-7b-3p, miR-3682-3p, miR-337-5p, miR-3200-5p, let-7b-5p, miR-412-5p, miR-4467 and miR-548 were noted to have the most significant $p$ value attributes. Using hierarchical clustering analysis, let-7b-5p, let-7b-3p and miR-3200-3p were identified as the most differentially expressed miRNAs between the favourable and unfavourable outcome groups (Fig. 2). KEGG pathway analysis of all 134 miRNAs identified multiple pathways involving a number of miRNAs with $p$ values < 0.05 (Fisher's exact test) after applying FDR correction. Since NE is associated with cell death, the apoptotic pathways were studied[4]. KEGG analysis identified three apoptotic pathways, namely the Hippo signalling pathway, P53 signalling pathway and Lysine degradation pathway. However, the only apoptotic pathway to differentially express both let-7b-3p and let-7b-5p between the Group 1 neonates with favourable and unfavourable outcome was the Hippo signalling pathway. The use of reverse pathway analysis of the Hippo pathway confirmed let-7b-5p to be in the top 10 most significant miRNAs of all the 771 miRNAs involved in this pathway. Using Normfinder, miR-454-5p was identified as a suitable endogenous control from 2795 miRNAs detected in the NGS analysis, with a good stability value of 0.68. Following NGS data analysis, let-7b-3p, let-7b-5p and miR-3200-3p were studied for validation as candidate miRNAs along with miR-454-5p as endogenous control in the next step using Reverse Transcription and quantitative Polymerase Chain Reaction (RT-qPCR).

### Validation of control and candidate miRNAs through RT-qPCR.

Independent validation of the endogenous control and three candidate miRNAs was performed on the three neonatal subgroups (1 A, 1B and 2; $n = 15$ each) after excluding umbilical cord blood samples. Neonates in Group 1 (moderate to severe NE with TH) had a significantly lower Apgar score at 10 min, were more likely to require respiratory support and need chest compressions compared to those in Group 2 (mild NE without TH) as shown in Table 1. There was a higher proportion of male neonates in Group 1, particularly in the unfavourable outcome group when compared to Group 2, although this was not statistically significant. We have shown that there was a strong correlation between the MRI outcome and two-year neurodevelopmental outcome in this cohort[16]. Table 1 shows that most neonates in unfavourable MRI group had substantial degree of brain injury with higher scores in all the regions, namely basal ganglia and thalamus, posterior limb of the internal capsule, white matter and cerebral cortex according to the scoring system used[17]. Of the neonates with moderate to severe NE treated with TH, all those in the favourable MRI group had a normal neurodevelopmental outcome as opposed to only one in the unfavourable MRI group at a median age of 2.5 years. Eleven children in the unfavourable MRI group developed cerebral palsy.

MiR-454-5p was validated as a reliable endogenous control miRNA with stable expression across all groups (Fig. 3a) and time points (Sample 1 (S1)—Day 1, Sample 2 (S2)—Day 2–3 and Sample 3 (S3)—Day 5) (Fig. 3b) (Supplementary Data 1). Let-7b-5p

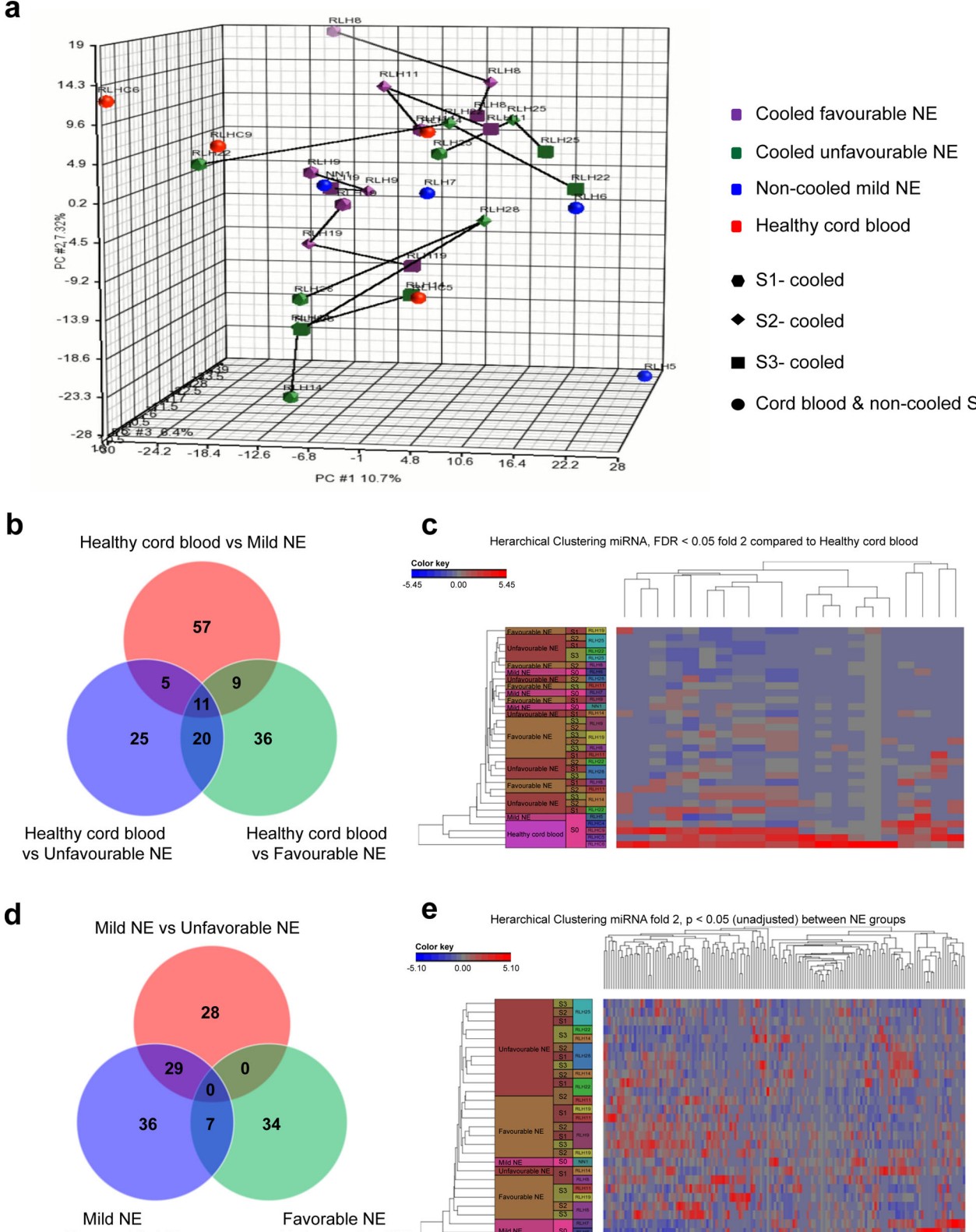

**Fig. 1 Identification of candidate miRNA through miRNA next-generation sequencing. a** PCA plot of all neonates examined with miRNA NGS according to their groups (Group 1A: moderate to severe NE with TH and favourable outcome (purple hexagon; S1, purple diamond; S2, purple square; S3), Group 1B: moderate to severe NE with TH and unfavourable outcome (green hexagon; S1, green diamond; S2, green square; S3), Group 2: mild NE without TH (blue circle), and Group 3: umbilical cord blood of healthy neonates (red circles). **b** Venn diagram showing the number of significant differential expression of miRNA between all three groups with umbilical cord blood. **c** Heatmap of all neonates in miRNA NGS with 17 significant miRNAs. **d** Venn diagram showing the differential expression of miRNA between all three groups excluding umbilical cord blood. **e** Heatmap of all neonates in miRNA NGS excluding umbilical cord blood showing no significant miRNAs.

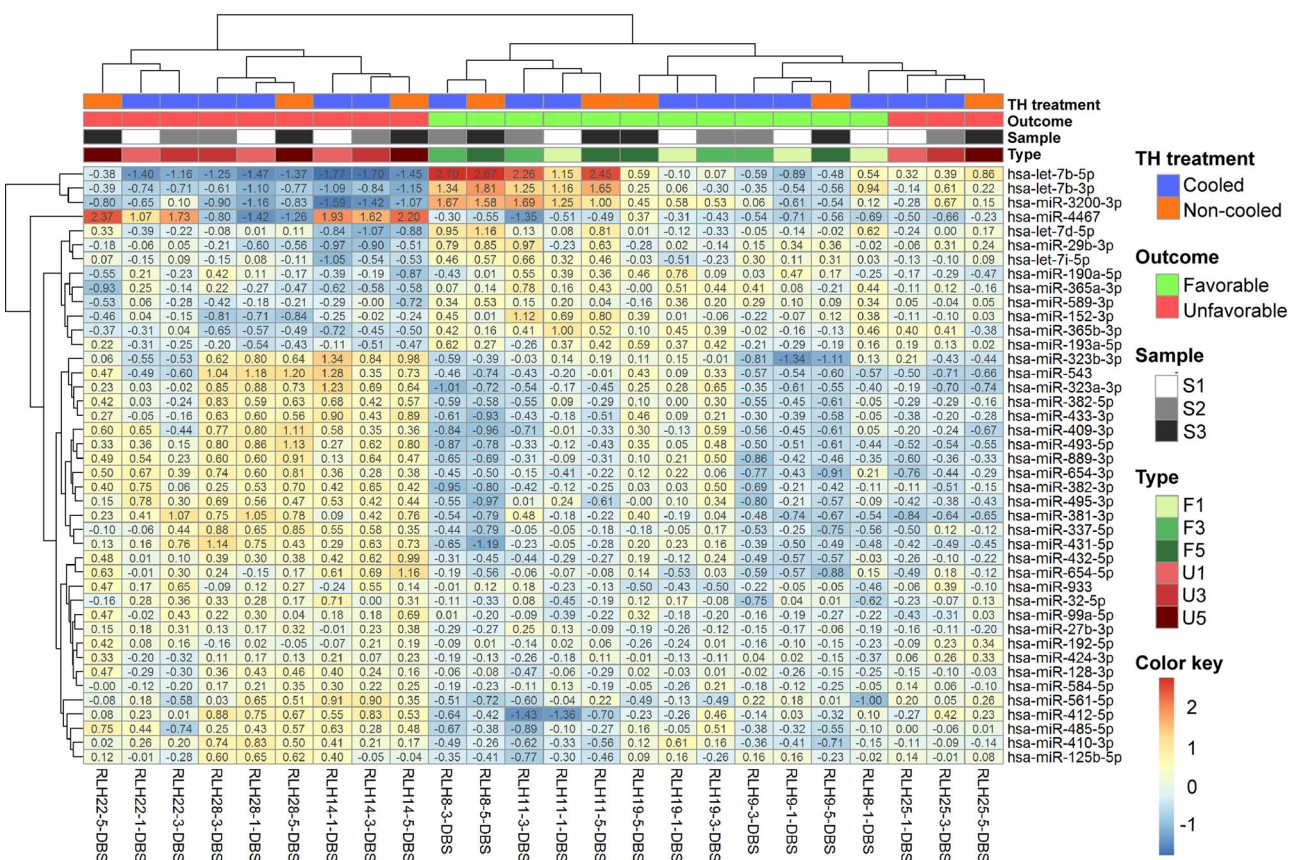

**Fig. 2 Hierarchical clustering analysis of all the differentially expressed miRNAs by outcome.** Hierarchical clustering analysis using the R program showing let-7b in the forms of 3p and 5p strands to be the most significantly changed miRNA in the moderate to severe NE neonates with TH treatment based on outcome. F1, F3 and F5 represent favourable outcome on days 1, 3 and 5 respectively; while U1, U3 and U5 represent unfavourable outcome on days 1, 3 and 5 respectively.

expression in S1 samples exhibited a significant increase in moderate to severe NE with TH and unfavourable outcome compared to mild NE without TH ($p = 0.019$) (Fig. 3c), while there was no significant change in temporal expression in Group 1 neonates between the outcome groups or other time points (Fig. 3d). However, let-7b-3p and miR-3200-3p did not demonstrate any differential expression both at S1 between all 3 groups or on temporal expression data at different time points on S1, S2 and S3 samples (Fig. 3e–h) (Supplementary Data 1). There was no statistical difference in the expression of all three candidate miRNAs between males and females within the groups. Because of the significant differential expression of let-7b-5p in neonates with NE, it was further studied in vitro and in two animal models of NE to further understand the mechanistic pathway.

**Validation of neuronal apoptosis in glucose-deprived in vitro cell cultures.** In order to study the apoptotic Hippo pathway, we first studied cleaved caspase-3 expression in the neuronal cells of an in vitro primary mixed cell cultures using glucose deprivation to mimic metabolic stress. There was significant increase in (mean + / SEM) cleaved caspase-3 expression in the neuronal cells of glucose-deprived cell cultures ($79.75 +/− 2.06$) when compared to control cell cultures ($53.58 +/− 0.74$) (Fig. 4a, Fig. 4b) (Supplementary Data 2).

**Reduced expression of let-7b-5p and YAP in glucose-deprived neuronal cells in vitro.** Let-7b-5p expression in neurones was studied in the in vitro NE model following confirmation of

apoptosis in the cell cultures under glucose-deprived conditions. In comparison with control cell cultures, glucose-deprived cultures showed significantly reduced expression (mean + /− SEM) of total let-7b-5p immunostaining area ($13.1 +/− 1.6$ vs. $5.6 ± 0.6$ A.U., $p = 0.004$) (Fig. 5a, Fig. 5b). More specifically, let-7b-5p expression in NeuN positive neuronal cells was significantly reduced ($50.78 +/− 1.36$ vs. $11.23 +/− 1.10$ % area, $p < 0.001$) (Fig. 5c) as was the percentage of neuronal let-7b-5p positive cells ($99.96 +/− 0.04$ vs. $79.94 +/− 8.31$, $p = 0.03$), confirming that the majority of let-7b-5p positive cells was neuronal (Fig. 5d) (Supplementary Data 2).

YAP, a core component of the Hippo signalling pathway, was studied in neuronal cells in the in vitro NE model as the KEGG analysis highlighted the significance of the apoptotic Hippo pathway in neonates with NE. Because a reduction in nuclear to cytoplasmic YAP expression is important to demonstrate an activated Hippo pathway,[18] co-immunostaining of YAP and NeuN, with Hoechst staining of the nucleus, was carried out to identify the neuronal (cytoplasmic) and nuclear YAP expression separately, as clearly observed (Fig. 5e). Neuronal /nuclear YAP ratio (mean + /− SEM) in glucose-deprived neuronal cells was significantly higher ($2.32 +/− 0.07$ vs. $1.67 +/− 0.08$, $p < 0.001$) when compared to control neuronal cells (Fig. 5f). This significant change in ratio was due to both a significant reduction (mean + /− SEM) in nuclear YAP expression ($21.21 +/− 0.81$ vs. $28.39 +/− 0.73$, $p < 0.001$) (Fig. 5g) and a significant increase in neuronal cytoplasmic YAP expression ($50.1 +/− 1.84$ vs. $44.16 +/− 1.07$, $p = 0.007$) (Fig. 5h) in glucose-deprived neuronal cells compared to neuronal cells in normal control

**Table 1 Clinical characteristics of patient samples analysed by RT-qPCR for the validation cohort.**

| Perinatal characteristics | Moderate to severe NE with TH and favourable MRI outcome (Group 1A) n = 15 | Moderate to severe NE with TH and unfavourable MRI outcome (Group 1B) n = 15 | Mild NE without TH (Group 2) n = 15 | P value |
|---|---|---|---|---|
| Gestational age, (completed weeks + days) | 40 + 4 (39 + 3- 41 + 4) | 40 + 2 (38 + 4-41) | 40 + 2 (39 + 3-41 + 4) | 0.430 |
| Male sex, n (%) | 10 (67%) | 12 (80%) | 6 (40%) | 0.071 |
| Birth weight (g) | 3700 (3318–3905) | 3240 (2745–3685) | 3520 (3145–3972) | 0.101 |
| Apgar score at 10 min | 5 (4–7) | 4 (4–6) | 9 (8–0) | <0.001*** |
| Worst pH within 1 h | 6.88 (6.77–6.95) | 6.87 (6.64–7.00) | N/A | 0.923 |
| Worst base deficit within 1 h | −17.45 (−14.12 to −20.85) | −19.3 (−17 to −22.9) | N/A | 0.381 |
| Need for respiratory support at 10 min, n (%) | 12 (80%) | 13 (87%) | 4 (27%) | 0.001** |
| Need for chest compressions, n (%) | 2 (13%) | 6 (40%) | 1 (7%) | 0.038* |
| Antenatal sentinel event present, n (%) | 3 (20%) | 1 (7%) | N/A | 0.598 |
| Pattern of MRI injury, n (score) | | | | |
| BG (Score 0–3) | 13 (0), 2 (1), 0 (2), 0 (3) | 2 (0), 0 (1), 5 (2), 8 (3) | N/A | N/A |
| PLIC (Score 0–2) | 15 (0), 0 (1), 0 (2) | 2 (0), 3 (1), 10 (2), | | |
| WM/CC (Score 0–3) | 11 (0), 3 (1), 1 (2), 0 (3) | 3 (0), 4 (1), 2 (2), 6 (3) | | |
| Normal neurodevelopmental outcome, n (%) | 15 (100%) | 1 (7%) | N/A | <0.001*** |
| Age at DBS sample S1 (decimal hours) | 14.8 (10.8–21.1) | 18.1 (12.4–21.1) | 22.8 (14.9–32) | 0.07 |
| Age at DBS sample S2 (decimal hours) | 60.2 (51.1–67.0) | 54.6 (51.1–61.4) | N/A | 0.907 |
| Age at DBS sample S3 (decimal hours) | 97.8 (88.9–104.1) | 95.4 (87.0–100.4) | N/A | 0.643 |

Values are median (IQR) unless indicated otherwise; N/A, Not Applicable; *, **, *** denote significant p < 0.05, p < 0.01, p < 0.001, respectively. *NE* neonatal encephalopathy, *TH* therapeutic hypothermia, *DBS* dried blood spot. MRI were rated using a validated method[17], with ranges of component scores for each of *BG* Basal Ganglia, *PLIC* Posterior Limb of the Internal Capsule, *WM* White Matter, *CC* Cerebral Cortex. Neonates with an unfavourable outcome had a severe pattern of injury including reversed or abnormal signal intensity bilaterally on T1- and/or T2-weighted sequences in the posterior limb of the internal capsule (PLIC); multifocal or widespread abnormal signal intensity in the basal ganglia and thalami (BGT); and severe widespread white matter (WM) lesions including infarction, haemorrhage and long T1 and T2. Neonates with MRIs predictive of a favourable outcome had either normal images or less severe patterns of injury that are associated with normal or only mildly abnormal neurodevelopmental outcomes. Consensus was reached in cases of disagreement. In this cohort we have shown that using this method, cerebral MRI is highly predictive of neurodevelopmental outcome[16].

conditions (Supplementary Data 2). These findings suggest activation of the Hippo pathway leading to increased neuronal YAP and reduced nuclear YAP, associated with reduced let-7b-5p expression under metabolic stress.

**Neuronal cell death in the cerebral cortex of rat neonatal encephalopathy models.** Two animal models of NE were studied as these may correspond to clinical NE in the human setting more closely. In the cerebral cortex, cleaved caspase-3 expression representing apoptotic cell death was significantly increased (mean +/− SEM) in hypoxic-ischaemic pups on the ipsilateral side (45.9 +/− 0.4, p = 0.004), contralateral side (51.9 +/− 3.6, p < 0.001) and the intrauterine inflammatory model pups (42.4 +/− 1.1, p = 0.006) when compared to control pups (30.0 +/− 3.4) (Fig. 6a, Fig. 6b). However, the total number of neuronal cleaved caspase-3 positive cells was not significantly different between the animal models (p value = 0.37, Fig. 6c) (Supplementary Data 3).

**Let-7b-5p expression in the rat neonatal encephalopathy models.** After confirming the presence of apoptosis in the NE models, let-7b-5p expression was studied in the cerebral cortex of both NE models in comparison to the control pups. Combined fluorescent in situ hybridisation (FISH) and immunohistochemistry (IHC) analysis using NeuN showed that let-7b-5p was predominantly expressed in cerebral cortical neurones (Fig. 7a). In the cerebral cortex, the percentage of neuronal let-7b-5p expression (mean +/− SEM) in comparison to the control pups was significantly reduced in both ipsilateral (69.9 +/− 2.3, p < 0.001) and contralateral (74.0 +/− 2.8, p < 0.001) sides in the hypoxic-ischaemic model and the intrauterine inflammation model (71.4 +/− 5.8, p < 0.001) (Fig. 7b) (Supplementary Data 3).

As for the human neonatal samples, let-7b-5p and let-7b-3p expressions were studied in the DBS of peripheral blood of NE animal models to understand the peripheral expression pattern. The relative quantification of let-7b-5p normalised to miR-454-5p in the peripheral blood of rat pups showed that there was a significant decrease in expression (mean +/− SEM) in the intrauterine inflammation model (0.28 +/− 0.09, p = 0.022), but not hypoxic-ischaemic model (0.90 +/− 0.27, p = 0.86) when compared to the control (1.0 +/− 0.17) (Fig. 7c). However, the relative expression of let-7b-3p normalised to miR-454-5p (mean +/− SEM) in hypoxic-ischaemic model (1.89 +/− 0.43) and intrauterine inflammation model (1.89 +/− 1.24) were not significantly different to the control (1.31 +/− 0.46) (overall p value = 0.90) (Fig. 7d) (Supplementary Data 3). This suggests that there was an alteration only in peripheral let-7b-5p expression after intrauterine inflammation. Similar to the in vitro NE model, let-7b-5p was confirmed to be differentially expressed both in peripheral blood and cerebral cortical tissue in the NE animal models when compared to control pups.

**Alteration in YAP expression of Hippo pathway in the rat neonatal encephalopathy models.** To understand the relationship between let-7b-5p and the Hippo pathway in the in vivo models, YAP expression was studied in the cerebral cortex of both the NE animal models in comparison to the control pups (Fig. 8a) as for the cell cultures using co-staining between the NeuN with YAP (neuronal, Fig. 8b) and YAP with Hoechst (nuclear, Fig. 8c).

In the cerebral cortex, there was a significant increase in the ratio of neuronal to nuclear YAP in NE animal models when compared to control pups (Fig. 8d). The neuronal/nuclear YAP ratio was significantly higher (mean +/− SEM) in hypoxic-ischaemic pups, ipsilateral side (0.92 +/− 0.01, p < 0.001) and contralateral side (1.0 +/− 0.05, p < 0.001),

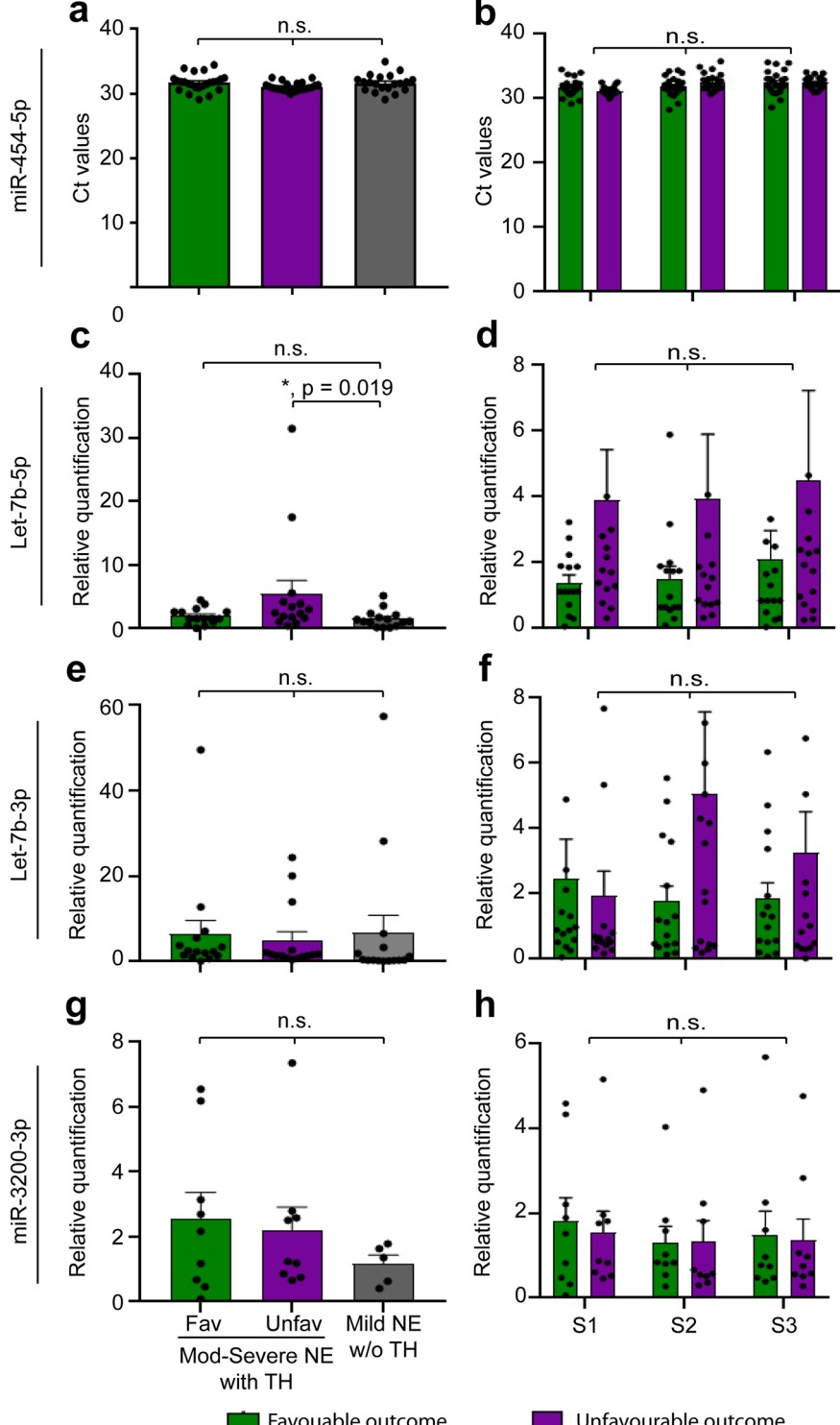

**Fig. 3 Expression of control and candidate miRNAs in neonates with encephalopathy. a, c, e** and **g** show the expression of miR-454-5p, let-7b-5p, let-7b-3p and miR-3200-3p respectively in all groups at S1 samples. Similarly graphs **b, d, f** and **h** show the temporal expression of miR-454-5p, let-7b-5p, let-7b-3p and miR-3200-3p, respectively, at various time points (S1, S2 and S3 samples) in neonates with moderate to severe NE with TH and favourable and unfavourable outcome. Green bars represent favourable outcome, while purple bars represent unfavourable outcome. Neonates with mild NE not receiving TH were denoted in grey bars. Relative quantification denotes $2^{-\Delta\Delta Ct}$ of the candidate miRNAs in comparison to endogenous control miRNA, miR-454-5p. Column graphs showing mean with error bars representing SEM. ns, not significant; * denote $p < 0.05$ using Kruskal–Wallis test for nonparametric data. Each group consists of $n = 15$ neonates. NE neonatal encephalopathy, TH therapeutic hypothermia, Fav favourable outcome, Unfav unfavourable outcome.

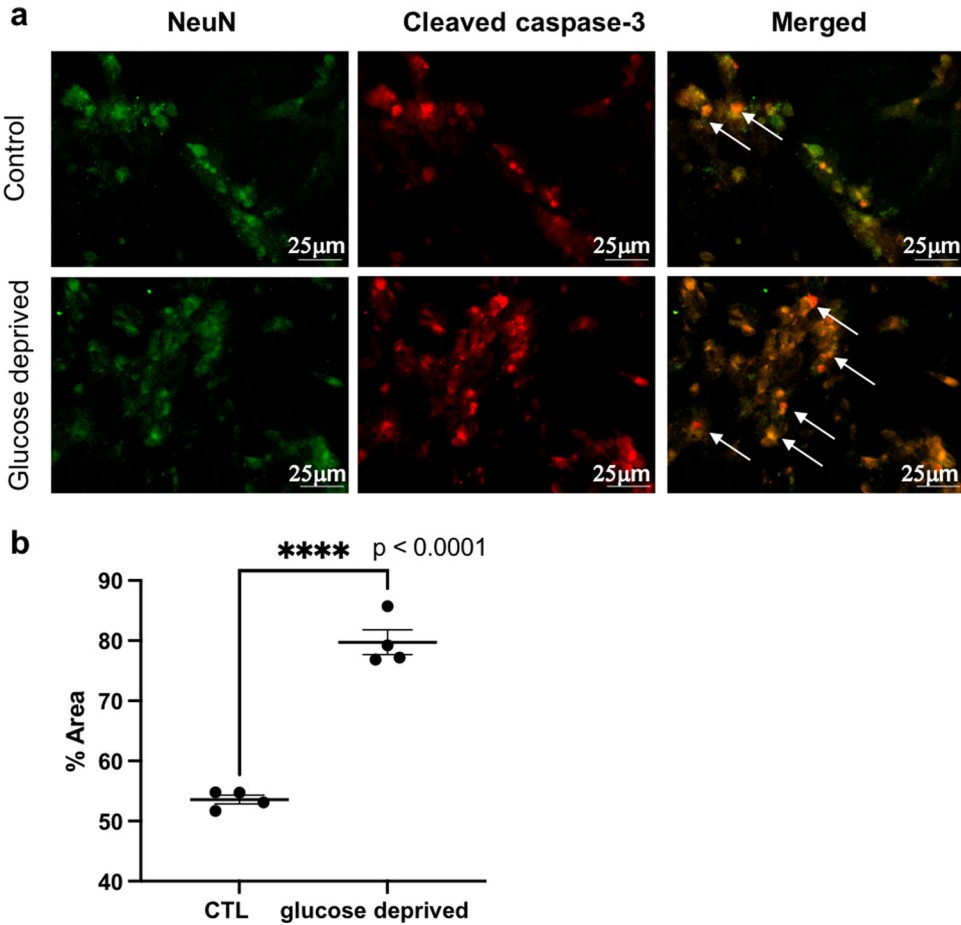

**Fig. 4 Neuronal apoptosis in normal and glucose-deprived neuronal cells in vitro. a** Images of NeuN positive neuronal cells (green) with cleaved caspase-3 (red) and coexpression is shown (yellow) in the control and glucose-deprived group. Arrows indicate coexpression. **b** Graph shows the significant increase in neuronal cleaved caspase-3 expression after glucose-derivation compared to the control group. Error bars represent mean with SEM. $n = 4$. **** denote significant $p < 0.0001$ using unpaired $t$ test. Scale bar = 25 μm. CTL control.

and in intrauterine inflammatory model pups (0.95 +/− 0.01, $p < 0.001$) when compared to control pups (0.74 +/− 0.02), indicating a higher proportion of neuronal cytoplasmic YAP to nuclear YAP in the cerebral cortex of NE models (Fig. 8d). This neuronal/nuclear YAP ratio received contribution from the cytoplasmic YAP expression that was significantly higher (mean +/− SEM) in the hypoxic-ischaemic ipsilateral side (40.5 +/− 3.7, $p = 0.002$), hypoxic-ischaemic contralateral side (39.5 +/− 3.4, $p = 0.003$) and intrauterine inflammatory model pups (38.3 +/− 2.3, $p = 0.001$) when compared to control pups (25.8 +/− 0.63, overall $p$ value = 0.0007) (Fig. 8e). The nuclear YAP staining was not significantly different in expression across both the NE animal modes ($p = 0.26$) compared to control pups. (Fig. 8f) (Supplementary Data 3). These results further confirmed that activation of the Hippo pathway in the neurones of cerebral cortex in both NE models leads to apoptosis.

## Discussion

While miRNAs have been noted to be involved in a number of pathophysiological processes in the central nervous system, our understanding of their role in neonates with moderate to severe NE is limited[4]. This study has used unbiased miRNA NGS to identify candidate miRNAs from DBS in neonates that may be associated with moderate to severe NE. Bioinformatic analysis

yielded let-7b-5p as a potential candidate and miR-454-5p as an endogenous control miRNA. Furthermore, for the first time, we report the association of let-7b-5p with the apoptotic Hippo pathway in the context of NE, following validation of let-7b-5p in an independent cohort of neonates with NE. To enable the study of the cellular and molecular mechanism associated with let-7b-5p and the Hippo pathway in NE, in vitro and in vivo models of NE were used. In vitro, glucose-deprived neuronal cells mimicking metabolic stress in NE showed a decrease in let-7b-5p, an increase in cytoplasmic YAP expression and a reduction in nuclear YAP expression. In animal models of NE, there was a significant decrease in let-7b-5p expression in peripheral blood of the rat intrauterine inflammation injury model and in the cerebral cortical neuronal cells of both the rat perinatal injury models. This was associated with a significant increase in neuronal cleaved caspase-3 expression and cytoplasmic YAP expression in the cerebral cortex of both animal models, indicating neuronal apoptosis linked to activation of the Hippo pathway.

Whilst little is known about let-7b-5p in NE, let-7b has been observed to be expressed in the mammalian brain and has been studied in a number of neurological conditions such as ischaemic stroke[19], glioma[20] and dementia[21]. Let-7b has also been shown to regulate neural stem cell proliferation and differentiation[22,23]. Additionally, extracellular let-7b may be a strong activator for Toll-like-Receptor signalling of the inflammatory pathway involved in adult neuronal cell death[24]. In the present study, let-

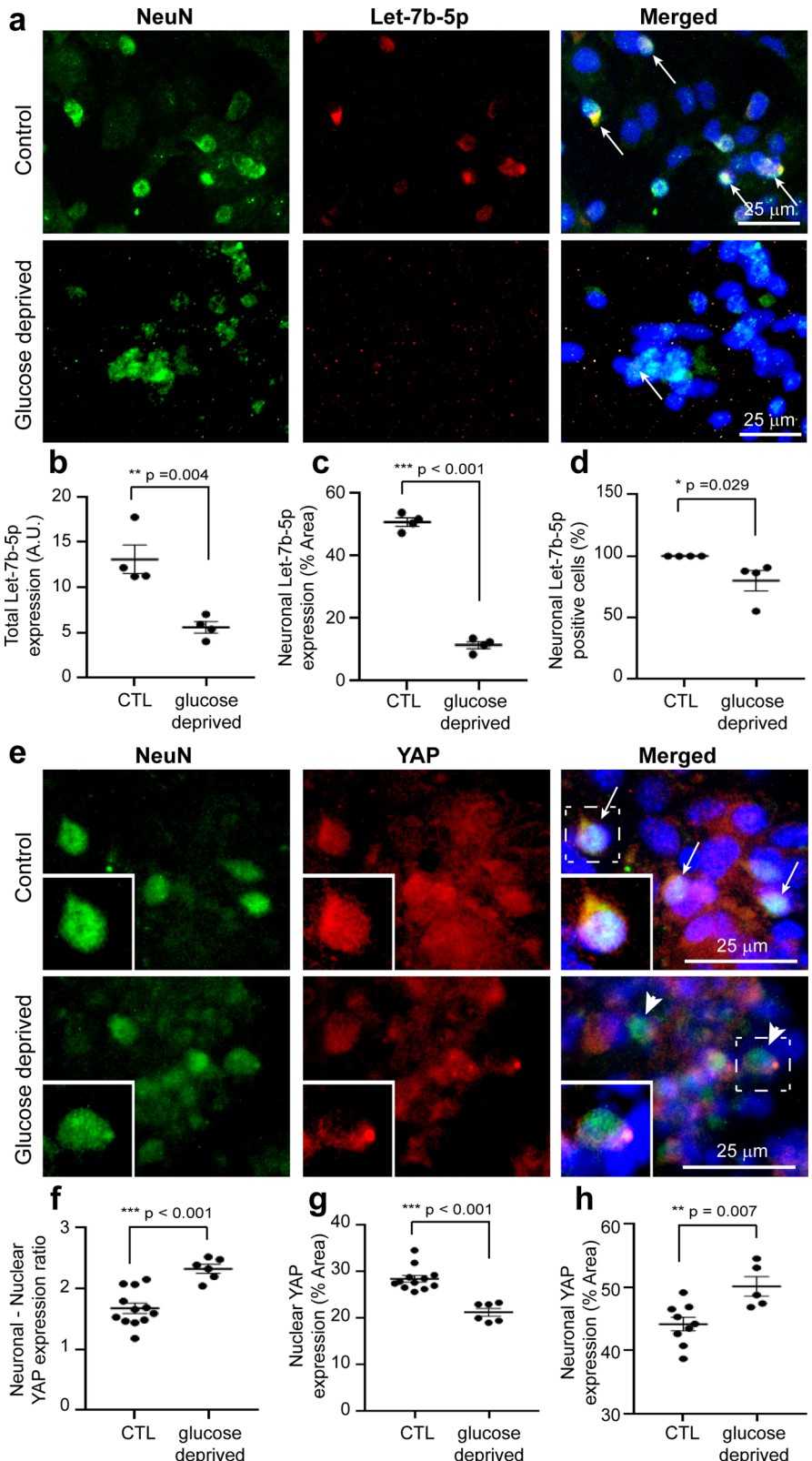

7b-5p was predominantly expressed in neuronal cells in both the NE animal models, with significant reduction in expression in the cerebral cortex of pups with hypoxic-ischaemic injury and intrauterine inflammation when compared to control pups.

Interestingly, we also noted a significant decrease in let-7b-5p expression in the peripheral blood of pups of the intrauterine inflammation model, but not the hypoxic-ischaemic injury

model. One possible explanation for this discrepancy may be related to the time period from the insult given; the hypoxic-ischaemic pups were euthanised at 24 h postinsult, while intrauterine inflammation pups had samples taken 9 days postinsult, thus allowing more time for changes in brain miRNA to be observed peripherally. This could also be due to an additional role of let-7b in neuroinflammation acting through other pathways.

**Fig. 5 Neuronal expression of let-7b-5p and YAP in normal and glucose-deprived neuronal cells in vitro. a** Images of NeuN positive neuronal cells (green) with let-7b-5p (red) and coexpression is shown (yellow) in the control and glucose-deprived group. Arrows indicate coexpression. **b–d** Graphs showing significant decrease in total let-7b-5p (**b**), neuronal let-7b-5p (**c**) and the percentage of neuronal let-7b-5p (**d**) after glucose-derivation compared to the control group. **e** Images of NeuN positive neuronal cells (green) with YAP (red) and coexpression is shown (yellow) in the control (arrows indicate cytoplasm expression) and glucose-deprived group (arrowheads indicate nuclear expression). **f–h** Graphs showing significant increase in neuronal/nuclear ratio (**f**), a significant decrease in nuclear YAP expression (**g**) and a significant increase in neuronal YAP expression (**h**) after glucose-derivation compared to the control group. Error bars represent mean with SEM. $n = 4$–12. *, **, *** denote significant $p < 0.05$, $p < 0.01$, $p < 0.001$, respectively, using unpaired $t$ test. Scale bar = 25 μm. CTL control.

Given the limited knowledge about the expression of let-7b-5p in NE, the use of two different animal models with different injury patterns allowed us to explore the let-7b-5p expression at the cellular level. Cleaved caspase-3 positive apoptotic neuronal cell death was significantly higher in the cerebral cortex of both the NE models when compared to control pups. This is in agreement with other studies showing activation of caspase-3 in neonatal hypoxic-ischaemic models both in rodents[25,26] and piglets[27]. The corresponding reduction in let-7b-5p and increase in cleaved caspase-3 expression highlights the potential role of let-7b-5p in the neuronal apoptotic cell death pathway[28].

Let-7b regulation has been noted to involve a number of targets and pathways, including vascular endothelial growth factor[29], hypoxia-inducible factor[30] and neuroinflammation[24,31], which are likely to be involved in the pathophysiology of NE. However, to date, let-7b-5p has not been studied in clinical or animal models of NE. Due to the presence of neuronal apoptosis in our models, we studied the role of let-7b-5p in the Hippo pathway, which was highlighted as one of the main apoptotic pathways through KEGG analysis.

The main function of the Hippo pathway depends on the translocation of YAP from the cytoplasm into the nucleus. Recently, a reduction in nuclear YAP protein levels has been correlated to reduced survival in Huntington's disease[32]. Although both the Hippo pathway and let-7b have been well studied independently in a number of oncological conditions, their role in NE remains to be explored in detail. In general, nuclear YAP is a potent growth promoter, and let-7b has a well-known tumour suppressor role[33]. We speculate that the interaction between the nuclear YAP and let-7b-5p may be inverse in order to maintain a normal environment with adequate cell growth /differentiation through a complex feedback system involving a number of antiapoptotic genes and possibly other miRNAs. In NE, the downregulation of let-7b-5p and the reduction in nuclear YAP could result in apoptotic neuronal cell death.

Currently, knowledge about specific interactions between the Hippo pathway and let-7b-5p is limited. A number of miRNAs have been noted to both positively and negatively regulate the Hippo pathway upstream[34]. On the other hand, the Hippo pathway can also regulate let-7b either directly through Dicer expression[35], indirectly through Lin28 which is influenced by phosphorylated Merlin, an upstream tumour suppressor in the Hippo pathway[34,36] or by p72 binding to the microprocessor complex needed in the biogenesis of miRNAs[37]. Some of these mechanisms also provide an association between the Hippo pathway and biogenesis of other miRNAs, with or without involving let-7b expression based on cell contact signals[35,37]. There are complex feedback loops between various miRNAs and the Hippo pathway, effectively controlling the expression of nuclear YAP, which in turn regulates antiapoptotic genes and thus controls cell growth and survival. Whilst our study has highlighted the changes in expression of both the let-7b-5p and the YAP protein in the Hippo pathway for the first time in

in vitro and in vivo NE models, the exact mechanism of the interaction between the two needs to be explored further.

Our study has a number of strengths. We have used neonatal samples from well-defined groups of babies with NE as a starting point to identify differentially expressed miRNAs through unbiased next-generation sequencing and further validated the candidate miRNA in a larger independent cohort of neonates using RT-qPCR. To the best of our knowledge, we have demonstrated a novel mechanism for let-7b in NE associated with the YAP protein of the Hippo pathway using both the in vitro and in vivo models.

Studies using umbilical cord blood as a biosample source to identify potential early biomarkers to stratify neonates with NE have been reported[38–40]. However, using miRNA NGS, we have also shown that umbilical cord blood has a different miRNA profile compared to that of neonatal blood and thus is not suitable as a control biosample for neonatal blood samples from later time points. Finally, in this study, we have validated our original technique of using DBS for RT-qPCR[8,9] to additionally perform miRNA NGS analysis. This would improve the feasibility of performing future large-scale studies in the neonatal population.

The main limitations of our study are the relatively smaller number of samples in each subgroup for various grades of NE in the miRNA NGS study and the individual variations in the sample collection times for the sequential blood samples. This might explain the lack of clear identification of significant miRNAs differentially expressed between the groups of neonates with varying degrees of NE. While the number of samples used provided the necessary power for statistical analysis, neonates with encephalopathy are known to have a varying clinical presentation that evolves with time. Therefore, a larger sample size in the subgroups at precisely set time points would have been more beneficial. We tried to reduce the clinical variations by selecting the neonates in each subgroup using clear categorisation as described in Supplementary Table 1. Additionally, as a novel study on the mechanism of action of a miRNA, we used both the peripheral blood and brain tissue of the animal models along with cell cultures to study the changes seen in NE at a cellular level. However, our work was done on P7 rat pups instead of P10 pups. Although P7 rat pups have been extensively used in NE models, they relate more to late preterm neonates of 32 to 34 weeks' gestation instead of full-term neonates[41,42]. Whilst our technique was robust, the findings may not be fully transferable to human neonates due to known variation in the expression of miRNAs in different species and tissues. However, of all miRNAs, let-7b is known to be an extremely well-conserved miRNA across various species, including humans and rodents.

In summary, through miRNA NGS analysis using dried blood spots of neonates with NE, and both in vitro and in vivo models of NE, this study highlights the potential importance of let-7b-5p and YAP in the Hippo pathway, in apoptotic neuronal cell death in NE. The complex interactions of let-7b-5p with its other targets and miRNAs warrant further investigation of its role in the pathophysiology of NE.

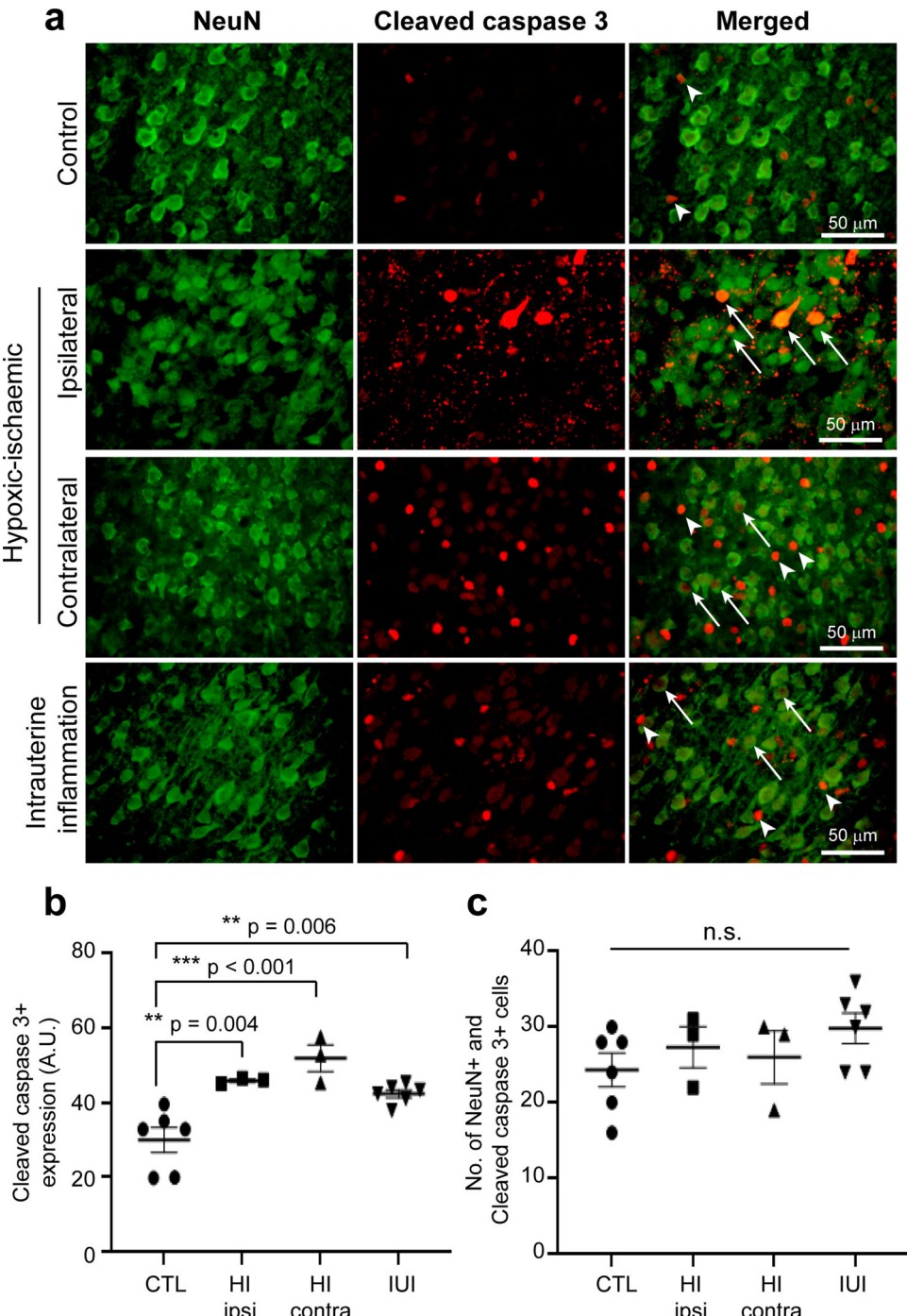

**Fig. 6 Neuronal apoptosis in the cerebral cortex of the rat neonatal encephalopathy models. a** Images of neuronal cells (NeuN positive, green) that can undergo apoptosis (cleaved caspase-3 positive, red) and coexpression is shown (yellow) in the cerebral cortex of control (arrowheads indicate non-neuronal apoptotic staining), hypoxic-ischaemic in the ipsilateral (arrows indicate neuronal apoptotic staining) and contralateral region (arrows indicate neuronal apoptotic staining), and intrauterine inflammation model (arrows and arrowheads indicate neuronal and non-neuronal apoptotic staining, respectively). **b, c** Graphs (**b**) show the statistical significance in cleaved caspase-3 apoptosis and (**c**) show no statistical significance in neuronal cell counts co-expressing cleaved caspase-3. Error bars represent mean with SEM ($n = 3$–6 per group). ns not significant; ** and *** denote significant $p < 0.01$ and $p < 0.001$, respectively, using ANOVA with Dunnett's multiple comparisons test. Scale bar = 50 µm. CTL control, HI hypoxic-ischaemic, ipsi ispsilateral side, contra contralateral side, IUI intrauterine inflammation.

## Methods and materials

**Clinical cohort.** Between January 2014 to January 2016, neonates > 36 weeks' gestation were recruited as part of the BIBiNS (Brain Injury Biomarkers in NewbornS) study from five UK neonatal units: The Royal London Hospital (Barts Health NHS Trust), Homerton University Hospital NHS Foundation Trust, Ashford and St Peter's Hospitals NHS Foundation Trust, University Hospital Southampton NHS Foundation Trust, and Norfolk and Norwich University Hospitals NHS Foundation Trust. The study was approved by a UK research ethics

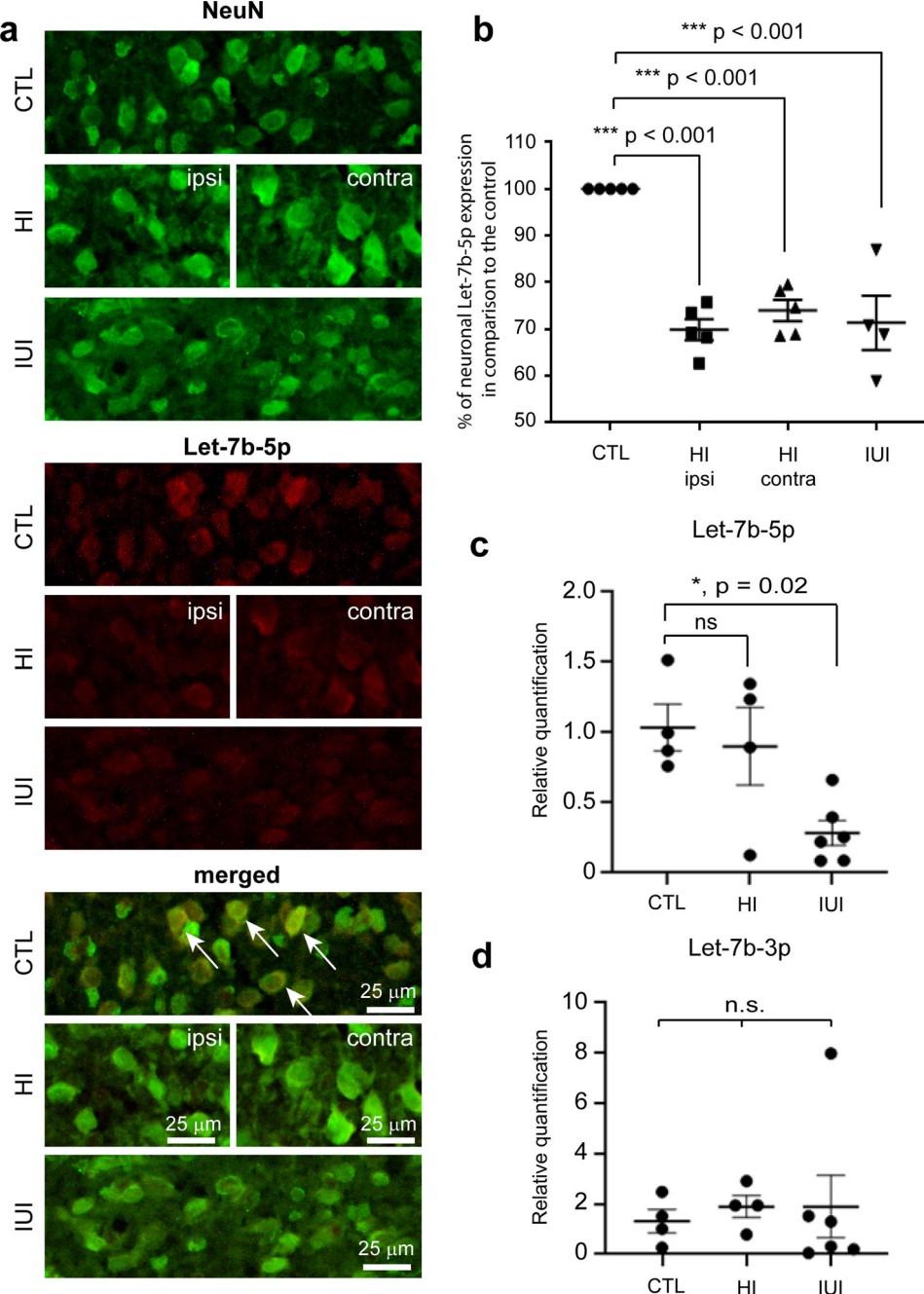

**Fig. 7 Expression of let-7b-5p in the cerebral cortex in all animal models. a** Images of NeuN positive neuronal cells (green) with let-7b-5p expression (red) and coexpression is shown (yellow) in the cerebral cortex of control (arrows indicate coexpression), hypoxic-ischaemic in the ipsilateral and contralateral region and intrauterine inflammation model. **b** Graph showing the statistical significance between the neonatal encephalopathy models vs. control groups ($n = 4$–$6$ per group). **c**–**d** Relative expression ($2^{-\Delta\Delta Ct}$) of let-7b-5p (**c**) and let-7b-3p (**d**) in peripheral blood in the form of DBS from the rat animal models. Error bars represent mean with SEM. ns not significant, * and *** denote significant $p < 0.05$ and $p < 0.001$, respectively using ANOVA with Dunnett's multiple comparisons test. Scale bar = 25 µm. CTL control, HI hypoxic-ischaemic, ipsi ipsilateral side, contra contralateral side, IUI intrauterine inflammation.

committee (London-Bromley, REC ref:13/LO/1738). Neonates were recruited with written consent from parents.

Neonates with a history of perinatal asphyxia including any one of the following: need for prolonged resuscitation beyond 10 min of birth; evidence of perinatal acidosis in any blood gas (umbilical cord or the baby's) in the first hour with pH < 7.00, base deficit > 16 mmol/L or high lactate; or an APGAR score of <5 at 10 min were recruited to the study. All eligible neonates were categorised prospectively within 6 h into three groups

based on their clinical presentation. Mild NE was defined as babies with evidence of perinatal acidosis or asphyxia but mild or no signs of encephalopathy. Babies with moderate to severe NE had signs of encephalopathy with altered consciousness, hypotonia, absent or reduced deep tendon reflexes, and/or seizures.

Group 1 included neonates with a clinical diagnosis of moderate to severe NE who fulfilled standard cooling criteria[43,44] and received TH for 72 h. Group 2 included neonates with mild

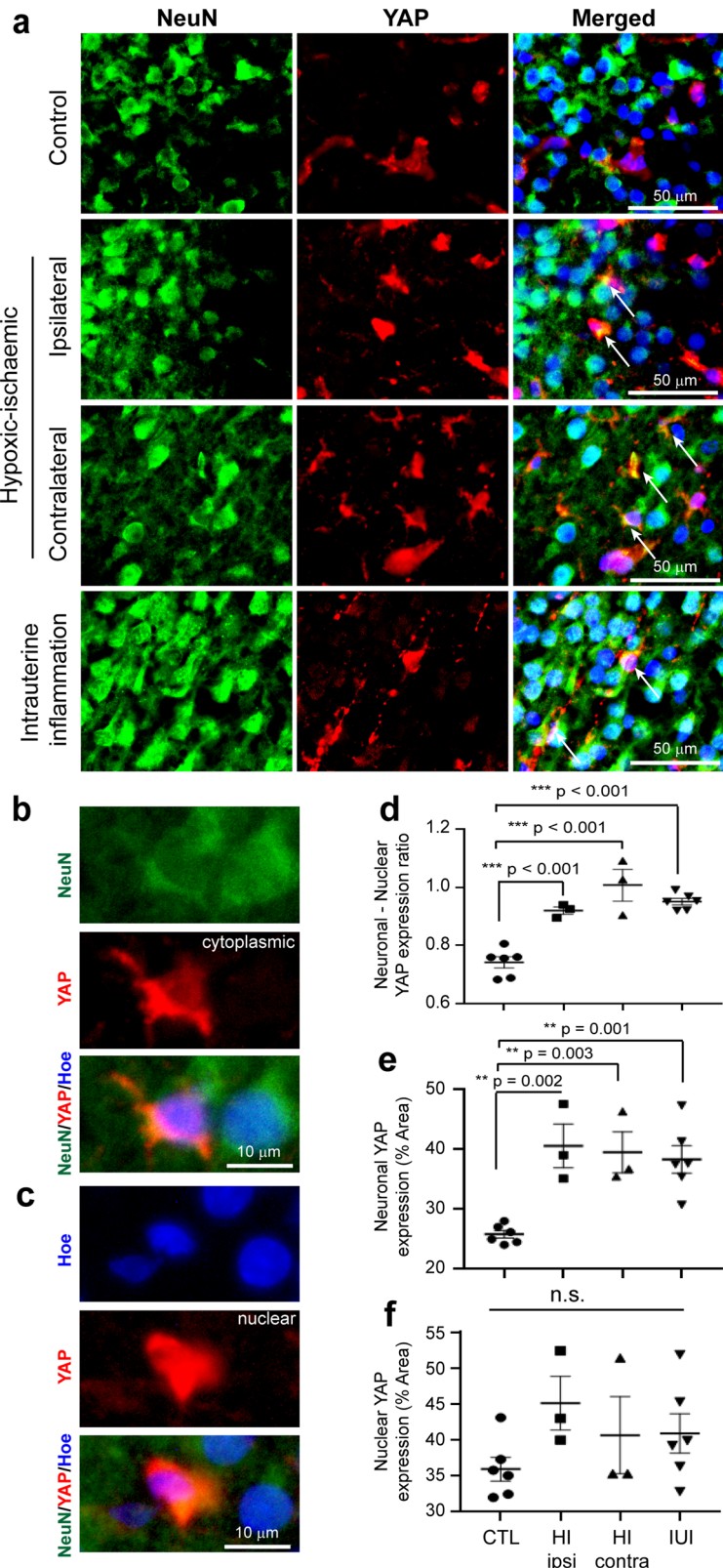

NE admitted to the neonatal unit with perinatal acidosis but did not satisfy criteria for TH treatment. Group 3 were cord blood samples from normal pregnancies. Short-term outcome was obtained for Group 1 neonates who received TH using conventional T1- and T2-weighted MRI sequences at 1.5 or 3.0 Tesla at the local centres according to unit protocol. Neonates in Group 1 treated with TH were further divided into 2 subgroups:

those with cerebral MRI predictive of either favourable (Group 1 A) or unfavourable outcome (Group 1B), based on the validated MRI scoring system[17] as previously described[16]. Additionally, long-term neurodevelopmental outcome was obtained and correlated with MRI outcome as detailed in the previous publication[16].

**Fig. 8 Neuronal expression of YAP in the cerebral cortex of the rat neonatal encephalopathy models. a** Images of NeuN positive neuronal cells (green) with YAP (red) and coexpression is shown (yellow) in the cerebral cortex of control, hypoxic-ischaemic in the ipsilateral (arrows indicate YAP positive neurones) and contralateral region (arrows indicate YAP positive neurones), and intrauterine inflammation model (arrows indicate YAP positive neurones). **b** Neuronal cytoplasmic YAP positive immunostaining is determined by strong coexpression of NeuN and YAP. **c** Nuclear YAP positive immunostaining is determined by strong coexpression of YAP with Hoechst staining in the nucleus. **d–f** Graphs showing a significant increase in the neuronal/nuclear ratio (**d**), significant increase in neuronal (**e**), but no changes in nuclear (**f**) YAP expression. Error bars represent mean with SEM. $n = 3–6$ per group. ns not significant; ** and *** denote significant $p < 0.01$ and $p < 0.001$, respectively, using ANOVA with Dunnett's multiple comparisons test. Scale bar = 50 μm for (**a**) and 10 μm for (**b, c**). CTL control, HI hypoxic-ischaemic, ipsi ispsilateral side, contra contralateral side, IUI intrauterine inflammation.

**Samples studied**. The neonates who underwent TH (Group 1) had samples collected at three-time points: sample 1 (S1) obtained on day 1 after achieving target body temperature of 33.5 °C; sample 2 (S2) obtained between days 2 and 3 during cooling therapy and prior to commencing rewarming; and sample 3 (S3) obtained on day 5 after rewarming was complete. Group 2 neonates had a single sample collected within 48 h of birth while Group 3 neonates had umbilical cord samples collected at birth. A drop of blood was collected at each sample point on an absorbent filter paper (Whatman 903 protein saver card) to form a DBS and stored in a polythene bag with a packet of desiccant at room temperature. Supplementary Fig. 1 shows the grouping of neonates and types of samples collected.

**MiRNA next-generation sequencing**. Based on the prediction of generating around 20 million reads per sample, a minimum sample size of 20 samples was estimated for miRNA NGS analysis. A cohort of 16 neonates was included, with four per subgroup selected on the basis of robust clinical grouping and MRI outcome to provide a total of 32 samples at the three-time points described above. The clinical characteristics of these neonates are described in Supplementary Table 1. With $2 \times 6$ mm DBS chad and previously published method[9] MiRNA NGS was performed using Illumina Nextseq 500 75 bp single-end high output run[45].

**Kyoto Encyclopaedia of Genes and genomes pathways analysis**. MiRNA pathway analysis was performed using mirPath V.3 to predict miRNA targets[46]. The Kyoto Encyclopaedia of Genes and Genomes (KEGG) pathways analysis was performed for three different comparisons through Tarbase V7.0 using pathway union. This included: 1) mild NE versus moderate to severe NE with favourable outcome (Supplementary Fig. 2), 2) mild NE versus moderate to severe NE with unfavourable outcome (Supplementary Fig. 3), and 3) moderate to severe NE with favourable outcome versus moderate to severe NE with unfavourable outcome (Supplementary Fig. 4). Reverse Pathway analysis of identified pathways was also performed for confirmation of the selected signalling pathway.

**Validation using quantitative Reverse Transcription PCR**. The TaqMan microRNA assays for quantitative RT-qPCR was performed according to manufacturer's protocols as previously described[9]. The three potential candidate miRNAs: let-7b-5p (Assay ID: 000378); let-7b-3p (Assay ID: 002404); and miR-3200-3p (Assay ID: 241643), plus the endogenous control miRNA miR-454-5p (Assay ID: 001996) were obtained from our miRNA NGS study. Validation by RT-qPCR was performed on both the miRNA NGS cohort and another larger independent cohort.

**Cell cultures**. Primary mixed cell cultures (containing approx. 27% neurones) from C57BL/6 mice pups at postnatal day 2 were carried out according to previously published work[47]. In order to study cell cultures under conditions of stress, glucose-deprivation was performed[48]. Cell cultures at a density of 6000 cells per well

in normal and glucose-deprived conditions were studied using combined FISH—IHC to identify the microRNA let-7b-5p expression in neurones using mouse anti-NeuN (1:500, MAB377, Merck Millipore)[49]. The cell cultures were double immunostained for YAP (rabbit anti-YAP, 1:100, 14074 S, Cell Signaling Technology) and NeuN, with nuclear Hoechst staining as previously described[50].

**Rodent models of perinatal brain injury**. All animal research was performed under the EU adopted Directive 2010/63/EU and reviewed by the local Comité National de Réflexion Ethique sur l'Expérimentation animale. The in vivo animal experiments for two established perinatal brain injury models for NE and control pups were carried out on time-pregnant Wistar rats purchased from CERJ (Le Genest, France). In the hypoxic-ischaemic model, pups of either sex at postnatal day 7 were anaesthetised using isoflurane and a unilateral common carotid artery ligation followed by hypoxia in a chamber was carried out as described before[51]. After 24 h, the pups were deeply anaesthetised, blood was collected onto absorbent filter paper as DBS with brains were dissected and stored in 20% sucrose at 4 °C until further processed. The intrauterine inflammation model involved an intraperitoneal injection of 300 μg/kg of lipopolysaccharide (E. coli, serotype 055:B5), to a pregnant rat at 20 days of gestation to produce inflammation, as previously described[52]. The blood and brain tissues were collected from pups on postnatal day 8 as described earlier. The control animals were from a group of naïve rat pups that underwent normal pregnancy conditions and were sacrificed on postnatal day 8 for blood and brain collection similar to experimental groups.

Ten μm brain sections between Bregma co-ordinates −2.20 to −0.20 mm containing the cerebral cortex, as shown in the postnatal rat brain atlas[53] were used in this study. The identification of the cell type that expressed let-7b-5p in the brain of rat animal models using mouse anti-NeuN (1:500, MAB377, Merck Millipore) for neurones was carried out using FISH and IHC, respectively. A similar protocol was used for double IHC using tyramide amplification for studying neuronal cell death in the animal models using rabbit anti-cleaved caspase-3 (1:100, Cat No. 9664, Cell Signaling Technology), and for expression of YAP in neurones using rabbit anti-YAP (1:200, Cat No. 14074 S, Cell Signaling Technology). Each animal model ($n = 3–6$) had 3–4 brain sections studied and imaged. In the hypoxic-ischaemic model, the injured side was defined as the ipsilateral side due to the more focal nature of the insult. In the intrauterine inflammation model, both left and right sides were combined together for analysis as the brain was affected globally by the endotoxin. Similarly, the uninjured control brain was analysed with combination of data from both the left and right side. Images were analysed using ImageJ (1.51 v) with a customised script.

**Statistics**. Bioinformatics analysis for miRNA NGS data was performed using Partek® Genomics Suite® software[54]. Heatmaps

and PCA were performed to identify differentially expressed miRNAs between the groups and at various time points. Normfinder was used to identify miRNAs with the best stability factor, to be suitable endogenous control miRNAs[55]. Relative quantification of the candidate miRNAs were performed using $2^{-\Delta\Delta Ct}$ method[56]. Hierarchical clustering was performed using R program[57]. All other data analysis was performed using GraphPad Prism 8. All statistical tests were 2-tailed with a significance level set with an alpha of 0.05. For continuous variables, statistical significance between three patient groups was compared using one-way ANOVA with Dunnett's post hoc test when comparing with the control group. For categorical variables, Pearson's chi-square and Fisher's Exact tests were used. For two-group comparisons, the Mann–Whitney $U$ test was used for nonparametric continuous variables while the unpaired $t$ test was used for parametric data.

**Reporting Summary**. Further information on research design is available in the Nature Research Reporting Summary linked to this article.

## Data availability

All miRNA next-generation sequencing data files are available from the Gene Expression Omnibus with the accession number(s) GSE181127. All other data are included in this published article (and its supplementary information files).

## Code availability

The customised plugins including 'Channel merging', 'Intensity and Co-localisation analysis' and "Cell count" macros used in ImageJ analysis to acquire data are available upon request.

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

## Acknowledgements

We are grateful to the neonates and the families who participated in BIBiNs study and thank the medical and nursing staff on all the neonatal units for their support. We would like to thank Nicola Openshaw-Lawrence, Nicky Holland, Vicky Payne and Karen Few for assistance with patient recruitment and data coordination at Royal London, Ashford & St Peter's, Southampton and Norwich hospitals respectively. We are also grateful for the work done by Dr Jane Evanson and Dr Olga Kapellou in reporting the MRI brain scans. We are very grateful to Barts Charity, London, UK for funding support for the study and to Research department at Ashford & St Peter's hospital for funding the Article processing charge for this publication.

## Author contributions

V.P., P.Y., P.G and D.S. contributed to the conception and design of the study; V.P., P.Y., R.I., M.M., P.C., E.W., C.M., L.S., A.B., P.C. and E.C. contributed to acquisition and analysis of the data; and V.P., A.M.T., P.G., P.Y. and D.S. contributed to drafting the text and preparing the figures.

## Competing interests

The authors declare no competing interests.
