## [Transparent Peer Review File · Communications Biology]

Reviewers' comments:

Reviewer #1 (Remarks to the Author):

Review

Brief summary of the manuscript

In this comprehensive study, the investigators first analyzed dried blood spots from a small sample of neonates with moderate to severe encephalopathy with favorable (Group 1A) or unfavorable outcomes (Group 1B) treated with hypothermia compared mild encephalopathy with no hypothermia (Group 2) and to a healthy group (Group 3). Their investigations led them to identify a number of microRNAs and particularly the microRNA let-7b-5p as a significant marker that separated groups using next generation sequencing and hierarchical clustering analysis. They then independently validated their findings in a slightly larger number from Group 1A, 1B and 2 (n=15 each group). They then went on to study the expression of let-7b-5p and YAP in a number of animal models including glucose-deprived neuronal cells, the vannucci model of HI and an inflammation model using LPS. They show a nice decrease in let-7b-5p in the glucose deprived cells. In the cerebral cortex of P7 rats, they show cell death and associated with decreased let-7b-5p. They also studied the expression of let-7b-5p in dried blood spots of the NE animals but only the inflammation model had a decrease. They then studied the Hippo pathway evaluating the cytoplasmic Yap to nuclear YAP in the cortex of the NE models with higher levels of YAP in the ischemic hemisphere and inflammatory model.

Overall impression of the work

This is a very nicely executed study showing that discrimination can be made between babies with favorable outcome treated with hypothermia vs those without favorable outcome with decreased expression of let-7b-5p in the unfavorable group. Outcome was tested at 2.5 years. They validated their findings in a slightly larger cohort and then tested in vitro and in vivo models that also showed the similar decline in the affected cells and animals and it was associated with YAP translocation suggesting involvement of the Hippo pathway. This is a very nice beginning to investigations into the role of the Hippo pathway and micro RNAs in NE.

Specific comments, with recommendations for addressing each comment

Specifically- more detail should be given regarding how the severity of encephalopathy was exactly determined in the three groups and when after birth. In addition, the outcome measures seemed to include both MRI and examinations. More detail should be given, especially in regard to the neurodevelopmental examinations. They do mention the scoring system for the MRI and it would be helpful to have a table showing the scores in Table 1 for both MRI and exams.

The numbers of patients are indeed limitations but the fact that they validated in a larger cohort lessens the concern. It is probably impossible because of numbers, but it would be nice to mention if there were sex differences given male predilection for worse outcome. A comment that the unfavorable outcome had a higher number of males would be important.

In regard to the in vitro studies, these are straightforward but they should show evidence of cell death by caspase like in the in vivo studies.

Unfortunately the in vivo studies were done in P7 rats which are more akin to late preterm humans than term. There next study should look at P10 and some mention of this limitation would be important. Also, it is not clear how many animals were used for these experiments. Were there sex differences here?

The data for the involvement of the Hippo pathway is very limited and not proof of principle so this

should also be mentioned in the limitations section.

Reviewer #2 (Remarks to the Author):

The authors found that microRNA let-7b-5p was differentially expressed in neonates with moderate to severe encephalopathy clinical samples and was associated with the apoptotic Hippo pathway. And then they detected the expression of let-7b-5p and YAP in glucose-deprived cell cultures as well as hypoxic-ischaemic and intrauterine inflammation rodent models. I was confused what the purpose of the authors. There are several questions need to be clarified.

1. Data provide in this manuscript did not supported the finding of the Hippo-YAP-let-7b. The authors did not demonstrate the association among let-7b-5p, YAP and Hippo pathway in this manuscript.

There is a miss link between let-7b-5p and YAP. The direct evidence of let-7b-5p regulating YAP stabilization and nuclear translocation need to be stressed.

2. The authors used both miR-454-5p and let-7b-3p as, please give an explanation.

3. X-axis of Fig.1 c is hardly to be readable.

4. P9L189: Figure 4 did not supported the findings that activation of the Hippo pathway leading to increased neuronal YAP and reduced nuclear YAP, leading to reduced le-7b-5p expression under metabolic stress.

5. P10L219: Was "let-7b-5p" mistakenly written as "let-7b-3p"? Let-7b-3p is decreased in the both the penumbra and blood of MCAO rat. So it is not an appropriate endogenous control miRNA.

6. P10L213-P10L225: the authors should check carefully this paragraph.

Reviewer #1	Responses to reviewer comments
1) Specifically- more detail should be given regarding how the severity of encephalopathy was exactly determined in the three groups and when after birth. In addition, the outcome measures seemed to include both MRI and examinations.	- We have amended the methods section to include a new paragraph with the clinical information about severity of NE and grouping of babies. We have also mentioned the time frame for inclusion in the study. (Page 18, Lines 389 - 397) - We have amended the 3rd paragraph in the same method section to detail both the short term MRI outcome and the long term neurodevelopmental outcome used in the study. (Pages 18-19, Lines 402 - 409)
2) More detail should be given, especially in regard to the neurodevelopmental examinations. They do mention the scoring system for the MRI and it would be helpful to have a table showing the scores in Table 1 for both MRI and exams.	- We have amended the methods section as mentioned above to provide information about additional long term neurodevelopment outcome used in the study. (Page 19, Lines 407 - 409) - The MRI scoring system validated by Rutherford et al* was used to categorise neonates with moderate to severe encephalopathy into favourable and unfavourable outcome. Table 1 has been amended now to include details on the scores for each pattern of injury seen in the MRI for neonates in Group 1. (Pages 29 - 30, Lines 702 - 719) - We have also included a sentence in the results section to highlight the significance of this to the readers. (Page 7 - 8, Lines 144 - 147) * Rutherford, M. et al. Assessment of brain tissue injury after moderate hypothermia in neonates with hypoxic-ischaemic encephalopathy: a nested substudy of a randomised controlled trial. Lancet Neurol. 9, 39-45, doi:10.1016/S1474-4422(09)70295-9 (2010).
3) The numbers of patients are indeed limitations but the fact that they validated in a larger cohort lessens the concern. It is probably impossible because of numbers, but it would be nice to mention if there	- We thank the reviewer for this valuable comment. We have included a sentence to inform the readers of the sex differences in the cohort in the results section. (Page 8, Lines 161 - 162)

were sex differences given male predilection for worse outcome. A comment that the unfavorable outcome had a higher number of males would be important.	- Additionally, we have added another sentence about the sex-specific results from the validation cohort. (Page 7, Lines 141 - 143)
4) In regard to the in vitro studies, these are straightforward but they should show evidence of cell death by caspase like in the in vivo studies.	- We have now performed additional lab work to study the cleaved caspase-3 expression in the cell cultures and included this in a new section (Pages 8 - 9, Lines 167 - 172) along with a new figure 4 (Page 36, Lines 786 - 796). - This has confirmed increased neuronal apoptosis in glucose deprived cell cultures when compared to control cultures. - We have made minor edits in the results section to amend the order of subsequent figures accordingly.
5) Unfortunately, the in vivo studies were done in P7 rats which are more akin to late preterm humans than term. There next study should look at P10 and some mention of this limitation would be important.	- We once again thank the reviewer for highlighting this limitation. We have included this with additional references 41 and 42 in our discussion section. (Page 17, Lines 364 - 367)
6) Also, it is not clear how many animals were used for these experiments. Were there sex differences here?	- We used 3-6 animals per group for all the experiments and this is mentioned already in the methods section titled 'Rodent models of perinatal brain injury'. (Page 22, Line 485) - Pups of either sex were used in the preparation of the models. This has also been mentioned in the methods section titled 'Rodent models of perinatal brain injury'. (Page 21, Line 464)
7) The data for the involvement of the Hippo pathway is very limited and not proof of principle so this should also be mentioned in the limitations section.	- We accept the reviewer's comment about lack of direct relationship between the Hippo pathway and let-7b-5p. In most places, we have only referred to the link between the Hippo pathway and let-7b as a likely association. - We have made this clearer by removing the two incidences of the phrase 'Hippo-

	YAP-Let-7b axis' from the abstract (Page 3, Line 53) and discussion sections (Page 16, Line 342) and replaced it with 'association between the Hippo pathway and let-7b'. - We have also added a comment in the discussion section, as requested by the reviewer, about this limitation. (Page 16, Lines 333 - 336)
--	---

Reviewer #2	Responses to reviewer comments
1. Data provide in this manuscript did not supported the finding of the Hippo-YAP-let-7b. The authors did not demonstrate the association among let-7b-5p, YAP and Hippo pathway in this manuscript. There is a miss link between let-7b-5p and YAP. The direct evidence of let-7b-5p regulating YAP stabilization and nuclear translocation need to be stressed.	- We accept the reviewer's comments on lack of direct link between let-7b and YAP from our research. This was also highlighted in the question 7 from reviewer 1. - We have taken on board comments from both the reviewers and edited the manuscript carefully to acknowledge this limitation in our work, as explained in the answers to the previous question. - We were unable to do additional lab work to show the direct relationship, due to limitations in funding and resources. We did not have access to more primary cell cultures for further detailed experiments. - Our work was primarily focussed on the bedside to bench approach of finding a candidate miRNA in neonates with NE and identifying the likely pathway involved. We acknowledge that downstream targets of the YAP protein influencing biogenesis of microRNAs, in particular let-7b, would be interesting to study for further knowledge. This would help to elucidate this novel association between the Hippo pathway and let-7b. However, this additional work would be another large piece of work as there is very little known about this association. - Therefore, we believe that our current study has provided new knowledge in the field of miRNAs in NE and highlighted areas for future work, in order to understand the mechanism involved.

2. The authors used both miR-454-5p and let-7b-3p as, please give an explanation.	- We believe that reviewer 2 meant in this comment that we had used both miR-454-5p and let-7b-3p as controls, as we had referred to the expression of let-7b-3p as negative control in Page 10, line 219 in original submission. - To clarify, we only used miR-454-5p as endogenous control for relative expression of all candidate miRNAs including let-7b-3p. We referred to the expression of let-7b-3p as not being significant when compared to let-7b-5p, and thus in itself acting as a negative control. We understand that this statement could be confusing, so we apologise for this, and we have now edited this original sentence to clarify our point clearly. (Page 11, Lines 222, 228 & 231)
3. X-axis of Fig.1 c is hardly to be readable.	- We have increased the resolution of this figure to ensure that the x-axis is clearly legible. (Page 31, Lines 720 - 723)
4. P9L189: Figure 4 did not supported the findings that activation of the Hippo pathway leading to increased neuronal YAP and reduced nuclear YAP, leading to reduced let-7b-5p expression under metabolic stress.	- As explained earlier in answer to Q1, we agree with the reviewer that the direct link between let-7b-5p and the Hippo pathway has not been studied in our work. So, we have rephrased this sentence correctly to: 'These findings suggest activation of the Hippo pathway leading to increased neuronal YAP and reduced nuclear YAP, associated with reduced let-7b-5p expression under metabolic stress.' (Page 10, Line 199)
5. P10L219: Was "let-7b-5p" mistakenly written as "let-7b-3p"? Let-7b-3p is decreased in the both the penumbra and blood of MCAO rat. So it is not an appropriate endogenous control miRNA.	- We did not use let-7b-3p as endogenous control. We apologise for the confusion caused by this sentence. We have amended this as explained in answer to Q2. (Page 11, Lines 222, 228 & 231)
6. P10L213-P10L225: the authors should check carefully this paragraph.	- We thank the reviewer for this comment. We have made minor corrections to ensure this paragraph reads clearly and correctly to avoid any misinterpretations. These changes are explained in our response to Q2 and Q5. (Page 11, Lines 222 - 234)

REVIEWERS' COMMENTS:

Reviewer #1 (Remarks to the Author):

All criticisms addressed

Reviewer #2 (Remarks to the Author):

Accept. I have no more questions.